# Lessons learned about willingness to adopt various protective measures during the early COVID-19 pandemic in three countries

**Ana Paula Santana** [1]*, **Lars Korn** [2,3,4], **Cornelia Betsch** [2,3,4], **Robert Böhm** [1,5,6]

**1** Department of Psychology, University of Copenhagen, Copenhagen, Denmark, **2** Media and Communication Science, University of Erfurt, Erfurt, Germany, **3** Centre for Empirical Research in Economics and Behavioral Sciences (CEREB), University of Erfurt, Erfurt, Germany, **4** Department of Implementation Research, Health Communication, Bernhard Nocht Institute, Hamburg, Germany, **5** Faculty of Psychology, University of Vienna, Vienna, Austria, **6** Copenhagen Center for Social Data Science (SODAS), University of Copenhagen, Copenhagen, Denmark

* apss@psy.ku.dk

## Abstract

### Background

Regarding the COVID-19 pandemic, concerted efforts have been invested in research to investigate and communicate the importance of complying with protective behaviors, such as handwashing and mask wearing. Protective measures vary in how effective they are in protecting the individual against infection, how much experience people have with them, whether they provide individual or societal protection, and how they are perceived on these dimensions.

### Methods

This study assessed the willingness to follow recommended measures, depending on these features, among participants from Germany ($n = 333$), Hong Kong ($n = 367$), and the U.S. ($n = 495$). From April 24th to May 1st, 2020, individuals completed an online survey that assessed the antecedents of interest.

### Results

It was shown that assumed effectiveness, previous experience, and intended self- and other-protection positively predicted willingness to comply across countries. When measures were mainly perceived as protecting others (vs. the self), individuals were less prone to adopt them. When a measure's effectiveness to protect the individual was perceived as lower, willingness to adopt the measure increased with higher levels of prior experience and collectivism. Moreover, protecting others was more strongly related to adoption when individuals had higher levels of collectivism and lower levels of individualism.

**Data Availability Statement:** Data and code for analyses are available from the Open Science Framework (https://osf.io/6drph/?view_only= eeae5576e5af4e17a6df9acd6ce1d8cd).

**Funding:** The work was funded by grants from the German Research Foundation (DFG, https://www.dfg.de/en/) to CB (BE 3970/8-2, BE 3970/11-1, BE 3970/12-1) and RB (BO-4466/2-2). The funders had no role in study design, data collection and analysis, decision to publish, or preparation of the manuscript.

## Conclusions

Emphasizing the benefit for others could be a means to lower the potential detrimental effects of low assumed effectiveness for individual protection.

## Introduction

Since the outbreak of the COVID-19 pandemic and the lack of effective treatments or scarcity of vaccines, health organisations and governments have strived to curb the disease spread by suggesting, requesting, or mandating non-pharmaceutical interventions (NPI), such as practicing social distancing or wearing a face mask. Their success in effectively reducing virus transmission depends on large-scale behavioral change [1].

Previous research indicates substantial interindividual variance within countries [2] and between countries [3] regarding the support for the imposed policies and adherence to the respective behavioral measures in response to the pandemic. The variance in adhering to the behavioral measures observed between individuals from the *same* country or cultural region may be captured by differences in people's subjective perceptions of the disease [4, 5], perceptions of trust in health organisations and governments [6], political attitudes [7], and personality traits [8]. Part of the variance observed *between* countries may be due to different governmental approaches in reacting to the pandemic. While some governments instituted lockdowns very early (e.g., China), others mainly advised on social distancing and hygiene practices (e.g., Sweden). In the latter, some people were still allowed to go to bars and cafes, whereas this behavior drastically diminished or was not possible in other countries. Moreover, people from different countries may vary in their previous experiences with certain measures (e.g., face masks) [9], which could affect their acceptance of such measures.

Another factor potentially contributing to between-country variation is cultural-psychological variables, such as collectivism and individualism [10]. For instance, people from countries with a larger valuation of collective welfare could be more likely to adopt measures that are (or are perceived as) not only serving individual health needs but also—or even primarily—public health needs due to the externalities they pose on others' health.

Although several studies have shown the influence of various factors on the acceptance of selected measures in a specific country, their independent and joint effects across various measures and countries are yet unclear. This study aimed to shed further light on this issue by (i) identifying adherence to seven different protective measures across three countries from Asia, Europe, and North America, using the same quota-representative sampling strategy across countries. Moreover, we investigated (ii) the independent and joint effects of psychological predictors on different adherence to protective measures. In doing so, we considered a comprehensive set of potential predictor variables, which might potentially account for variation within and between countries. In the next section, we introduce potentially important psychological antecedents for adhering to protective measures and develop our hypotheses accordingly.

### Perceptions of effectiveness and past experience with the measures

As different NPIs were simultaneously recommended across countries, it is quite challenging to assess the effect of each measure on curbing the virus spread [11]. For example, over the course of the pandemic, the effectiveness of masks has been heavily under debate [12, 13]. People may vary in how effective they perceive different measures due to such debates and respective media reports. Variance in the perceived effectiveness of actions against the pandemic

may also be attributed to cross-country differences [14]. Considering the present study was conducted in the beginning of the pandemic, there was likely more uncertainty surrounding the effectiveness of the protective behaviors. Moreover, perceived effectiveness is an important construct across different theories in health psychology; it is a component of the Protection Motivation Theory (PMT) and linked to core constructs of other frameworks, too (e.g., perceived benefits in the Health Belief Model) [15]. Studies using the PMT highlight that the perceived effectiveness of a health behavior to protect from a health threat (i.e., response efficacy) is associated with higher motivation to adopt behaviors such as vaccination or preventive measures against COVID-19 [16]. Therefore, from previous studies' findings [6, 17, 18], we expected that differences in individuals' perceptions regarding the effectiveness of different measures would also relate to their differential adherence to these measures. We hypothesized:

> H1: The willingness to adopt a specific protective measure is larger the more it is perceived as effective
>
> (effectiveness hypothesis).

Besides differences in the perception of protective behaviors' perceived effectiveness, people may also differ regarding their past experiences with these measures. The relationship between past experience and the uptake of preventive behaviors has been documented and explored by theoretical models in other contexts (e.g., natural hazards) [19, 20]. Concerning a situation as the COVID-19 pandemic, it might be easier to avoid handshakes and other physical forms of greetings if one is less accustomed with such manners. This can vary due to individual preferences and due to cultural norms. Regarding the latter, one study showed that the magnitude of the COVID-19 pandemic tended to be smaller in countries with a stronger handwashing culture [21].

It is documented that the frequency of having performed a certain behavior and resulting habit formation are associated with behavior change in different domains, such as oral hygiene and food safety [22]. Further evidence investigating habit formation suggests that, when making decisions, associations are made between the chosen option and the neglected one [23]. For instance, the adoption of a protective behavior in the past might have increased the value of the chosen option and devalued the not chosen one (i.e., not adopting the behavior). Thus, people might learn from experience that adopting a certain health behavior (e.g., wearing a mask) has a greater value than not doing it. Therefore, having experienced several specific behaviors often in the past might increase the chances that this behavior becomes a habit or at least easier to adopt in an emergency, such as the COVID-19 pandemic. This pattern can be assumed for mask wearing, one of the measures that helped contain disease transmission during the 2003 severe acute respiratory syndrome (SARS) epidemic in Hong Kong [24]. With the first cases of COVID-19 reported in late January 2020 in Hong Kong, wearing masks was rapidly recommended with high compliance [9], whereas the acceptance of and adherence to mask wearing in many Western countries resulted in somewhat more discussion and opposition (e.g., voluntary vs. mandatory mask policies [25, 26]. This study focuses on the direct experience individuals have had with behaviors regarding past frequency. Therefore, the second hypothesis is as follows:

> H2: The willingness to adopt a specific protective measure is larger the more experience an individual has had with this measure in the past
>
> (experience hypothesis).

## Intended self-vs. other-protection, collectivism, and individualism

Regarding the nature of the different recommended protective behaviors, some of them are considered primarily for self-protection (e.g., washing hands for 20 seconds) [27], whereas others are considered to mainly protect other people (e.g., wearing masks) [28]. However, there is often no strict distinction between mere self-protection measures and mere other-protection measures. For example, physical distancing can protect both oneself and others from an infection. While there are actual differences in which measures protect others vs. the self or both, individuals may vary in how they perceive the measure. Thus, one of the aspects we focus on is how much self- and other-protection people attribute to different measures.

Moreover, there are also individual differences regarding intended self- vs. other-protection that guide individual action. These primary motivations can be affected depending, for instance, on which benefit is publicly communicated or emphasized [29, 30]. Studies investigating how both pro-self and pro-social motives influence adherence to pandemic measures have contradictory findings. A manipulation of personal vs. public framings (i.e. addressing pro-self vs. pro-social motives, respectively) suggested that a public social framing was more effective in increasing the willingness to adopt protective measures [31]. However, a study assessing the protective value of behaviors for the self and for the public found that the perception of self-protection was more important in adhering to a specific measure [28].

People from different cultural backgrounds vary in how much they focus on benefits for oneself or other. These differences could moderate the effect of the intended self- or other-protection of measures on adherence to certain behavioral regulations. For instance, dimensions such as independence vs. interdependence and tightness vs. looseness influence how communities engage in collective efforts [32]. Cultures where individualism is highly endorsed (e.g. North America) are usually considered independent, whereas cultures strongly attached to collectivist values are considered interdependent [10]. The concepts of individualism and collectivism have been associated with specific types of pro-social behavioral tendencies [33]. Previous evidence found that individualism was positively linked to the tendency to help others get approval, whereas collectivism was associated with pro-social behaviors evoked by an emotional circumstance, in response to a request, or performed anonymously [33]. Such findings can be relevant when considering a disease outbreak scenario in which some protective measures often have a pro-social aspect (e.g., mask wearing).

Relatedly, societies with a high historical pathogen burden show a predominance of collectivistic [34] and tight cultural values [35], which can be attributed to the need for coordinating behavior to tackle such issues. One study investigating the timeframe from 1950–2008 found that infectious and zoonotic outbreaks were positively related to individualism, suggesting that individualistic countries were subjected to more disease outbreaks throughout the period investigated. It has been suggested that collectivistic behaviors can have a protective function, which may be activated when a specific threshold of compliance is achieved [34]. Thus, the dimensions of individualism and collectivism might affect individuals' primary motivation (pro-self vs. pro-social) to engage in collective efforts, such as following health authorities' behavioral recommendations.

This study investigates how collectivism and individualism, along with the intended self- and other-protection, affected people's willingness to adopt different protective measures during a specific timeframe within the COVID-19 pandemic. Although there are studies investigating the effects of pro-sociality and cultural aspects on compliance [29, 36], this study extends previous research by examining whether individualism and collectivism also drive individual differences in the underlying motivation for various measures. Specifically, the third hypothesis states the following:

*H3: The willingness to adopt protective measures is larger, as people perceive them as protecting others, particularly for individuals with higher levels of collectivism*

*(but not for individuals with higher levels of individualism; collectivism–pro-sociality hypothesis).*

Note that the numbering of the hypotheses reported here differs from the pre-registration, where they are numbered as H4, H5, and H6, respectively.

## Materials and methods

### Study population

The sample of participants ($N$ = 1,195; $M_{age}$ = 47.56, $SD$ = 17.41; 46.53% female) was recruited online between 04/24/2020 and 05/01/2020. Participants were from Germany ($n$ = 333), Hong Kong ($n$ = 367), and the United States (US) ($n$ = 495) and were members of an ISO 26362:2009-compliant online panel [37]. The external panel provider was responsible for recruitment and compensation of the participants. Due to problems in filling certain quotas, the U.S. sample was oversampled to reach the required quotas. Drop-out rates after quota filling were 7.92% for Germany, 11.16% for Hong Kong, and 15.05% for the United States. The aforementioned sample sizes only consider complete participation. Note that our aim of sampling participants from several countries was not to compare responses between countries, but rather to create sufficient variance in the predictors of interest, which have been shown to vary between countries and cultural contexts. Non-probability quota samples were used, representing the distribution of age × gender of the adult population of the respective countries (see S1 File for more details). Individuals between 18 and 74 years were eligible for participation. Participants were admitted to the survey or screened out on the first page based on quotas.

The present data came from an independent survey included in a larger study, which was an experiment investigating the intention to get vaccinated with a hypothetical vaccine. However, only data about the willingness to adopt protective measures were analyzed and presented here. Sample size was based on an *a priori* power analysis of unrelated hypothesized effects not reported in this paper. Yet, a sensitive power analysis for mixed-effect regressions using the 'simr' [38] package in R [39] revealed a power of 0.72 to detect a small effect size, and a power of $>$ .99 for a medium effect size (see details in S1 File). We did not exclude any participants from the analysis. Note that a question about having been infected or knowing someone who was infected with COVID-19 was preregistered as an exclusion criterion but was not assessed in the study due to technical issues.

### Procedure and measures

The average time to complete the study was 16.57 minutes ($SD$ = 81.6), and participants were remunerated by the panel provider. The survey was presented in English for participants from Hong Kong and the U.S., and in German for participants from Germany. In addition to the measures for this study, the complete survey also included measures of perceived risk, intolerance of uncertainty, social norms, trust in the government, and the economic impacts of getting ill with COVID-19 (see preregistration). All measures analyzed here were assessed in the order listed below.

**Individualism and collectivism.** These variables were assessed using the horizontal and vertical individualism and collectivism scale [10]. The 16 items were rated on a 7-point scale (1 = *never* to 7 = *always*) and presented randomly to the participants. We did not differentiate between the horizontal and vertical subtypes of the subscales. Thus, only two scores (i.e.,

individualism and collectivism) were computed by the average of items of each dimension (Cronbach's α was 0.73 for individualism and 0.84 for collectivism). Sample items are: "I'd rather depend on myself than others.", for the individualism subscale; and "The well-being of my co-workers is important to me.", for the collectivism subscale.

**Willingness to adopt protective measures.** This was the main dependent variable in this study. Participants rated on a 7-point scale: "What is the probability that you would take up the following protective measures?" (1 = *definitely not* to 7 = *definitely*). Seven protective measures were included in this study: washing hands for 20 seconds, covering the mouth and nose when coughing or sneezing, wearing a mask in public, taking any steps to avoid being near someone who has symptoms of the disease, avoiding physical contact with friends/family, avoiding handshakes, and keeping distance from other individuals (6 ft or 1,50 m) in the public. No aggregate index was computed for the dependent variable since mixed-effect regressions with random intercepts for participants were used in the analysis.

**Perceived effectiveness.** The item read "How effective do you consider each of the following protective measures to be?" and was answered on a 7-point scale (1 = *not effective at all* to 7 = *very effective*). Answers were given separately for each of the protective measures listed above.

**Intended self-protection vs. other-protection.** This section consisted of two parts. First, participants answered an item that read "I would adopt the following measures to protect *myself*" for all measures (see above). Then, they answered "I would adopt the following measures to protect *other people*" for each of the measures. Answers were given on a 7-point scale (1 = *fully disagree* to 7 = *fully agree*).

**Past experience with the measures.** Participants answered the item "In your everyday life or during another health situation, how often have you. . .", e.g., ". . .worn a face-mask," on a 7-point scale (1 = *never* to 7 = *often*), separately for each measure.

## Statistical analyses

Analyses were conducted in R [39] using the package *lme4* [40] for the mixed-effect regressions. A repeated measure design was considered with the willingness to adopt protective measures as the main dependent variable (seven values of willingness per participant, one for each measure). The predictors were past experience with the measure, intended self- vs. other-protection, perceived effectiveness of the measure, individualism, and collectivism. As for the dependent variable, no aggregate index was computed for predictors measured repeatedly. Considering this study was an addition to an experimental study, we also conducted additional analyses including the experimental condition as predictor. Although most predictors were measured after the experimental manipulation, it did not affect the measures. The corresponding analyses are reported in the S1 File (S3 and S4 Tables in S1 File).

To predict individuals' willingness to adopt protective measures, we conducted mixed-effect regressions, treating the participant as a random effect to account for the interindividual variance in adopting (different) protective measures [41]. Cross-country variance was accounted for by adding country as a dummy predictor (baseline: Germany). Note that with only three different countries, there are too few observations to allow robust estimations when treating country as a random effect [40].

## Results

### Preregistered analyses

Overall, the mean willingness to adopt the measures was high across countries (Germany: M = 6.03, SD = 0.99, Hong Kong: M = 5.60, SD = 0.92, U.S.: M = 6.10, SD = 1.03). Model 1

tests the hypotheses of the study, and Model 2 allows for interactions of perceived effectiveness and past experience with the other predictors (Table 1).

**Perceived effectiveness of and experience with measures.** Across all models, the results indicate that protective measures were more likely to be adopted: (i) when individuals perceived them as more effective in protecting against infection with COVID-19, and (ii) when individuals had prior experience with the respective measure, supporting both the effectiveness and the experience hypothesis. Regarding the intended self-vs. other-protection, participants reported a higher likelihood of adoption when they intended to protect themselves compared to when they intended to protect others. Moreover, higher levels of collectivism increased the likelihood of adopting protective measures, whereas the effect was the opposite for individualism. Lastly, participants from Hong Kong reported to be, in general, less likely to adopt protective measures compared to German participants (whereas German and U.S. participants did not differ).

There were also several significant interaction effects (Model 2). For individuals who perceived the measures as more effective, the effects of prior experience and collectivism on increasing willingness to adopt the measures were weaker compared to individuals who perceived the measures as less effective. This means that the detrimental effect of a measure's low(er) perceived effectiveness on adoption was partly compensated for when participants had prior experience with the measure or higher levels of collectivism. In contrast, intending to protect oneself and higher levels of individualism amplified the effect of perceived effectiveness on a measure's likelihood of adoption. Moreover, the analysis revealed that when people had experience with a measure, it did not matter that much whether they intended to protect themselves or others.

**Collectivism, individualism, and intended other-protection.** Next, we test the collectivism–pro-sociality hypothesis, i.e., the assumption that individuals scoring higher on collectivism (but not individuals scoring higher on individualism) are more likely to adopt a measure if they intend to protect others. We included the interaction terms of the intended other-protection with both individualism and collectivism to predict adoption motivation (Table 1). The interaction between collectivism and intended other-protection was positive and marginally significant in Model 1 and significant in Model 2. The interaction between individualism and intended-other protection was negative and significant. As shown in Fig 1, the more people intended to protect others, the higher the willingness to adopt a measure, and this effect was stronger at higher levels of collectivism and weaker at higher levels of individualism, respectively.

## Exploratory analyses

**Replicating models per country.** Our main aim in sampling participants from several countries was not to compare responses between countries, but rather to create sufficient variance in the predictors of interest, which have been shown to vary between countries and cultural contexts [3]. Nevertheless, running Models 1–2 separately for each country is insightful because it allows us to compare the observed effects across countries (S5 and S6 Tables in S1 File). Fig 2 displays the standardized-regression coefficients from Model 2 separately for each country. Compared with the first analysis (all countries), the preregistered main effects were consistent across countries, providing support for the generalization of the effectiveness and the experience hypothesis. Regarding the *collectivism–pro-sociality hypothesis*, the effect found in the preregistered analysis replicated in the German sample only (hypothesis was partially supported in Model 1, supported in Model 2). For the other countries, these effects were less consistent. Although the remaining interaction terms were generally less robust and varied

**Table 1. Mixed effects regressions models testing the pre-registered hypotheses (Model 1), and exploring interactions (Model 2).**

| Predictors | Model 1 | | | Model 2 | | |
|---|---|---|---|---|---|---|
| | β | CI | p | β | CI | P |
| Effectiveness | 0.23 | 0.21, 0.25 | <.001 | 0.22 | 0.20, 0.25 | <.001 |
| Experience | 0.11 | 0.09, 0.12 | <.001 | 0.13 | 0.11, 0.14 | <.001 |
| IOP | 0.28 | 0.25, 0.30 | <.001 | 0.26 | 0.24, 0.28 | <.001 |
| ISP | 0.25 | 0.23, 0.27 | <.001 | 0.22 | 0.20, 0.25 | <.001 |
| Collectivism | 0.06 | 0.03, 0.08 | <.001 | 0.07 | 0.05, 0.09 | <.001 |
| Individualism | −0.03 | −0.05, 0.00 | 0.025 | −0.03 | −0.06, −0.01 | 0.007 |
| Country: Hong Kong | −0.11 | −0.16, −0.05 | <.001 | −0.12 | −0.18, −0.07 | <.001 |
| Country: USA | 0.00 | −0.06, 0.05 | 0.853 | 0.00 | −0.05, 0.05 | 0.995 |
| IOP x Collectivism | 0.02 | 0.00, 0.03 | 0.058 | 0.03 | 0.01, 0.05 | 0.002 |
| IOP x Individualism | −0.02 | −0.04, −0.01 | 0.010 | −0.04 | −0.06, −0.02 | 0.001 |
| Experience x Effectiveness | | | | −0.03 | −0.05, −0.01 | 0.002 |
| IOP x Effectiveness | | | | −0.01 | −0.03, 0.00 | 0.128 |
| ISP x Effectiveness | | | | 0.04 | 0.02, 0.05 | <.001 |
| Collectivism x Effectiveness | | | | −0.04 | −0.06, −0.02 | <.001 |
| Individualism x Effectiveness | | | | 0.03 | 0.01, 0.05 | 0.004 |
| Experience x IOP | | | | −0.03 | −0.05, −0.01 | 0.001 |
| Experience x ISP | | | | −0.05 | −0.07, −0.03 | <.001 |
| Experience x Collectivism | | | | 0.02 | 0.00, 0.03 | 0.092 |
| Experience x Individualism | | | | 0.01 | −0.01, 0.03 | 0.476 |
| Random Effects | | | | | | |
| $\sigma^2$ | 0.68 | | | 0.66 | | |
| $\tau_{00}$ | 0.16 $_{ID}$ | | | 0.16 $_{ID}$ | | |
| ICC | 0.19 | | | 0.19 | | |
| N | 333 $_{ID}$ | | | 367 $_{ID}$ | | |
| Observations | 2331 | | | 2569 | | |
| Marginal $R^2$ / Conditional $R^2$ | 0.558 / 0.643 | | | 0.565/ 0.649 | | |

*Note*. Mixed effects model (prediction of willingness to adopt protective measures [1–7]): All predictors were centered at their mean. Standardized coefficients are reported here. IOP = intended other-protection, ISP = intended self-protection. Country coding: Germany was considered as the baseline. CIs indicate 95% confidence intervals.

between countries, the interaction effect between effectiveness and intended self-protection remained consistent.

**Perceived relative pro-social value of behavioral measures.** Previous analyses have shown that the intention to protect oneself and others increases the willingness to adopt protective behaviors. However, it might well be that certain measures are perceived as serving a greater pro-social benefit (compared to a pro-self benefit) than others. Consequently, the relative importance of wanting to protect oneself vs. others for a given behavior could potentially affect the motivation to adopt the respective behavior. To explore this issue, we computed a protection motivation index (PMI) for each measure by subtracting the reported intention to protect oneself from the reported intention to protect others. All PMI values exceeding zero were classified as pro-social, and all others were classified pro-self. Accordingly, pro-social protective measures are those that are perceived as having a greater pro-social benefit compared to pro-self benefit.

Using this new variable as a predictor under the otherwise same model specifications as above, we found further support for both the effectiveness and the experience hypothesis

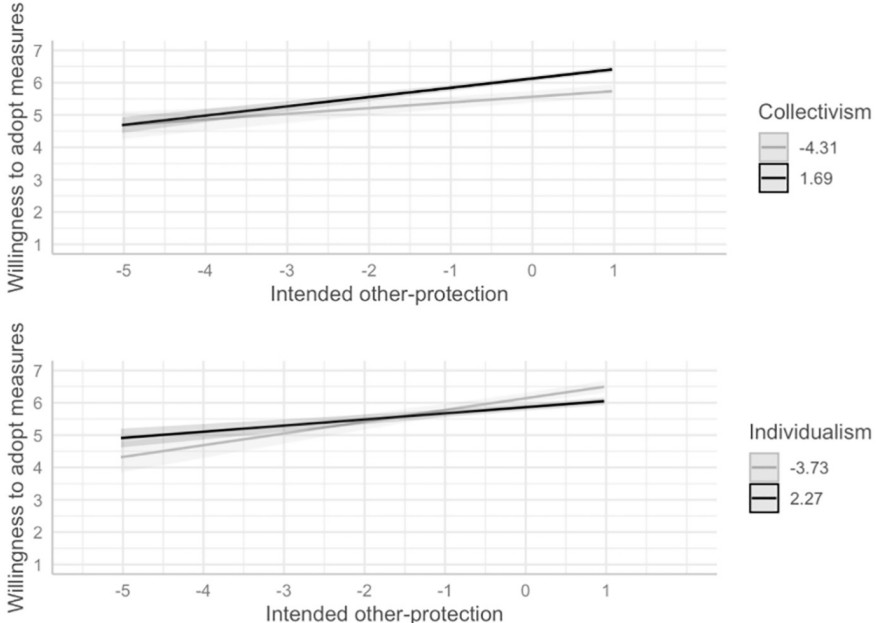

**Fig 1. Collectivism–pro-sociality hypothesis.** The figure displays predicted values and not the observed data. Lighter (vs. darker) shading for collectivism and individualism represents minimum and maximum values of the variables after mean-centering. Collectivism and individualism moderate the effect of pro-social motivation on the willingness to adopt protective measures. Independent variables were mean-centered. Darker (lighter) lines, shades, and the values in the legend indicate higher (lower) levels of the predictors.

([Table 2]). More importantly, the negative effect of PMI suggests that participants were less willing to adopt those measures that they perceived as serving mainly a pro-social benefit. Interactions of PMI with effectiveness and experience further indicate that (i) the perceived effectiveness of a measure was less important for adopting this measure when participants perceived a primarily pro-social benefit of doing so and (ii) a lack of experience with a certain measure was particularly harmful for adoption motivation when this measure was perceived as primarily pro-social in nature.

## Discussion

Considering the global scenario after the first wave of the COVID-19 pandemic, this study presents a cross-cultural assessment of a collection of preventive behaviors, suggesting learnings for future pandemic preparedness. We assessed the willingness to follow recommended measures depending on perceived effectiveness, past experience, intended self- and other-protection. Our main results confirmed that perceiving a measure as more effective and having experience employing such measures in the past predicted willingness to adopt it. Many studies conducted during the COVID-19 pandemic [3, 28] support the role of perceived effectiveness in adopting protective measures. While this association may seem rather straightforward, people's judgments of how effective the recommended measures are can be surrounded by uncertainty. For instance, the COVID-19 pandemic has shown that the effectiveness of NPIs can differ depending on aspects such as the intrinsic nature of the measure (e.g. physical distancing vs. cleaning of surfaces) and the time of implementation (e.g. earlier vs. later in the pandemic) [11]. Furthermore, subjective effectiveness might not always reflect the real efficacy of a certain implemented measure.

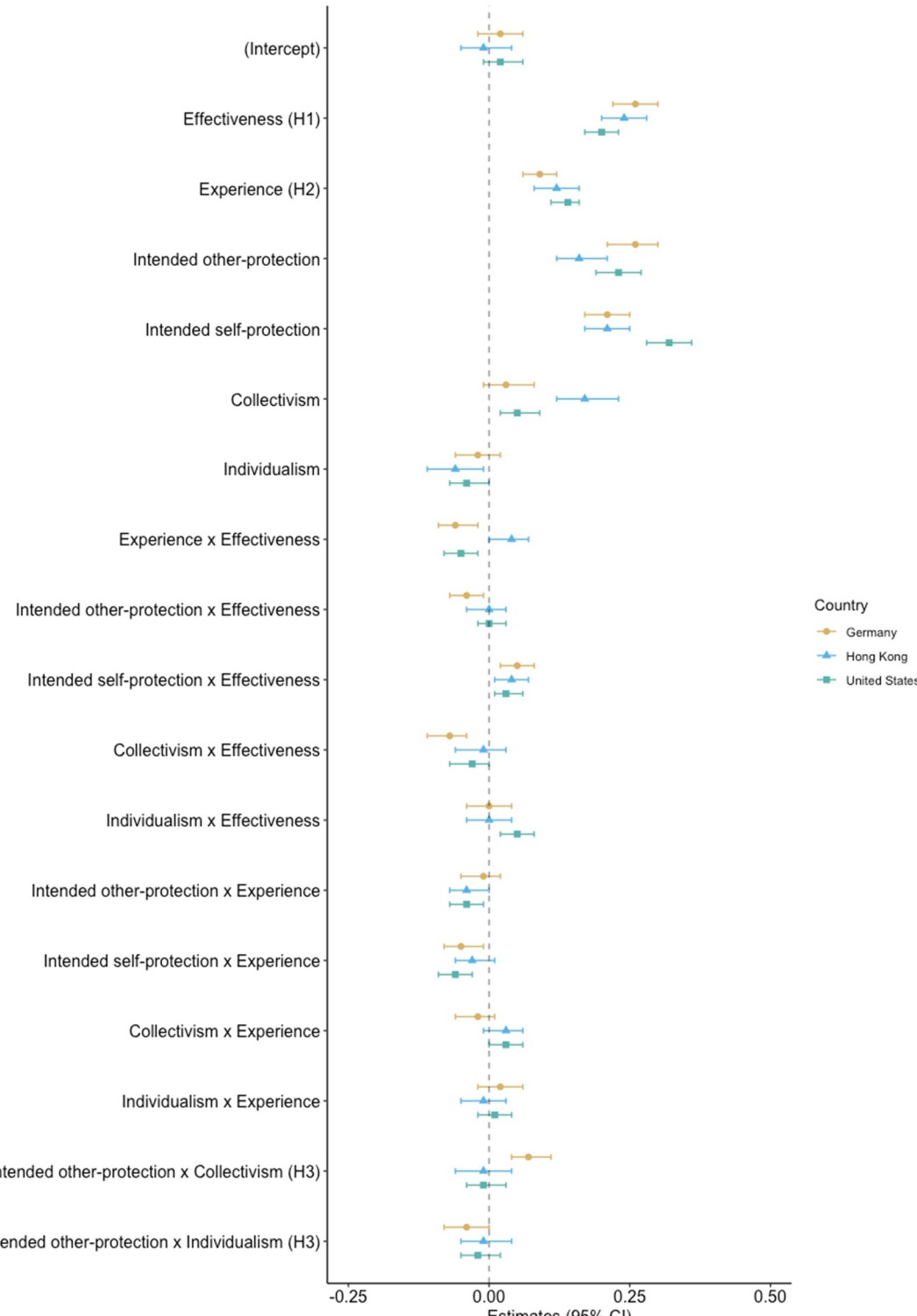

**Fig 2. Predictors of willingness to adopt the measures per country.** Standardized coefficients from Model 2 separately for each country. H1, H2, and H3 correspond to the three preregistered hypotheses.

**Table 2. Reproducing the models with the PMI.**

| Predictors | Model 3 | | | Model 4 | | |
|---|---|---|---|---|---|---|
| | β | CI | p | β | CI | p |
| Effectiveness | 0.46 | 0.45, 0.48 | <.001 | 0.43 | 0.41, 0.46 | <.001 |
| Experience | 0.17 | 0.15, 0.19 | <.001 | 0.17 | 0.15, 0.19 | <.001 |
| PMI[Pro-social] | −0.18 | −0.22, −0.14 | <.001 | −0.25 | −0.29, −0.21 | <.001 |
| Collectivism | 0.14 | 0.11, 0.17 | <.001 | 0.15 | 0.12, 0.18 | <.001 |
| Individualism | −0.05 | −0.08, −0.02 | 0.001 | −0.05 | −0.08, −0.02 | 0.001 |
| Country: Hong Kong | −0.16 | −0.23, −0.10 | <.001 | −0.19 | −0.26, −0.13 | <.001 |
| Country: USA | 0.01 | −0.05, 0.07 | 0.711 | 0.01 | −0.05, 0.07 | 0.768 |
| Experience x Effectiveness | | | | −0.14 | −0.15, −0.12 | <.001 |
| PMI[Pro-social] x Effectiveness | | | | −0.16 | −0.20, −0.12 | <.001 |
| Collectivism x Effectiveness | | | | −0.01 | −0.03, 0.01 | 0.172 |
| Individualism x Effectiveness | | | | 0.01 | −0.01, 0.03 | 0.541 |
| Experience x PMI[Pro-social] | | | | 0.04 | 0.00, 0.08 | 0.042 |
| Experience x Collectivism | | | | 0.01 | −0.01, 0.03 | 0.218 |
| Experience x Individualism | | | | 0.01 | −0.01, 0.03 | 0.566 |
| Random Effects | | | | | | |
| $\sigma^2$ | 0.80 | | | 0.76 | | |
| $\tau_{00}$ | 0.25 ID | | | 0.25 ID | | |
| ICC | 0.24 | | | 0.25 | | |
| N | 333 ID | | | 367 ID | | |
| Observations | 2331 | | | 2569 | | |
| Marginal $R^2$ / Conditional $R^2$ | 0.423 / 0.562 | | | 0.441/ 0.582 | | |

*Note.* PMI: for each measure, intention to protect oneself was subtracted from the intention to protect others. PMI values exceeding zero were classified as pro-social, and pro-self otherwise. Brackets indicate the category of the variable. CIs indicate 95% confidence intervals. Models 3–4 were also replicated per country (S6 and S7 Tables in S1 File).

This study contributes to such findings by demonstrating the role of both effectiveness and past experience when other variables are considered. For instance, the results suggest that perceiving a measure as less effective might be compensated for by higher levels of past experience and collectivism. This has implications for the communication of preventive behaviors guidelines, as emphasizing the protection of others, anchoring the behavior in prior experiences and activating collectivist values yield promising alternatives to highlighting only the effectiveness of the recommended behaviors. However, considering the exploratory analyses, this point must be interpreted with caution, as interactions vary across countries.

Moreover, the main analysis showed that the intention to protect oneself and others both predicted willingness to comply with the measures. Research on other health preventive behaviors supports this idea by showing that pro-social and individualistic interests can promote preventive behavior (e.g., vaccination) [42]. In this study, there was little variance between the participants' intentions to protect themselves and others. This might have compromised the disentangling role of each motivation in this context. Thus, the exploratory analysis categorizing the measures with a PMI was an attempt to further clarify these relationships. However, results from this exploratory analysis should be interpreted with caution, also because using difference scores as predictor variables is associated with some methodological problems [43].

Evidence involving the interplay between more pro-social vs. pro-self motivations during this pandemic are somewhat contradictory. This study found that participants were less willing

to adopt measures classified as pro-social (vs. pro-self). Accordingly, a previous study showed that people have a higher tendency to adopt measures considered as primarily self-protective rather than the ones regarded as mainly protecting others [28]. However, contrasting evidence suggests that messages with a pro-social (vs. individual) appeal have a higher effect on the willingness to follow protected measures [31]. In the case of apparent individualistic preference, one possible explanation could be that individuals' regard to other's condition is attenuated when a higher risk of getting infected is perceived [42]. Nevertheless, this interpretation must be cautiously considered in this study, as we did not analyze how people perceived risks.

Considering the variables investigated here, our results also offer insights into the relative importance of the predictors. Perceived effectiveness as well as intended self- vs. other-protection were the strongest predictors of the willingness to adopt the behaviors. This trend is also reflected in the models using the PMI. Given that these variables were of particular importance, it could be fruitful to target them when aiming to increase compliance with protective measures.

Notably, this study has some limitations, and the results presented here, particularly those from exploratory analyses, should be interpreted with caution and need further replication. For instance, the way pro-social concerns were operationalized and the rationale behind the PMI index is specific to the methodology adopted here. Therefore, comparisons with other studies investigating both pro-social and self-centered motivations can be challenging, as such constructs can be operationalized differently. Further limitations concern the assessment of the willingness to adopt protective measures. First, this assessment did not measure actual behavior, and we cannot be sure that the effects would be the same if that was the case. Nevertheless, evidence has shown that intentions and behavior often correlate to a moderate degree [44]. Second, our sample showed a high willingness to adopt the behaviors, which might have resulted in an underestimation of the effects found. For instance, in countries where compliance and regulations are weaker, or at times when compliance wanes due to habituation effects, individual differences are likely to matter more [45, 46]. Another limitation arises from the fact that we could not comply with one of the pre-registered exclusion criteria (i.e., whether one has been infected with COVID-19), and this might have affected our results. Overall, individuals who had experience with the disease, and depending on the type of experience (e.g., mild or severe), could have adapted their risk perceptions and health behaviors accordingly. For instance, if someone was infected and had only mild symptoms, they could have perceived the disease as posing less risk, subsequently reducing their protective behavior. As the data collection took part at a very early time during the pandemic, however, we assume that only a very small proportion might have had been infected at all. Finally, there were only small differences in the levels of collectivism and individualism between countries, and, surprisingly, the average level of collectivism (individualism) of participants from Hong Kong was descriptively below (above) the levels from German and U.S. participants (see S1 Table in S1 File). We refrain from speculations why this was the case but want to highlight that the analyses and their interpretations are based on individual-level responses instead.

Future studies should investigate and replicate the role of the predictors investigated here, especially effectiveness, experience, intended other-protection, and intended self-protection about actual adoption of the measures. Additionally, the relationship between effectiveness and past experience with other predictors might be tested in experimental contexts to see if some of the compensating effects observed here are replicated.

## Conclusion

In summary, the results reported here present some implications for communication aimed to increase compliance to protective health measures. Emphasizing or explaining the

effectiveness of protective measures can positively affect the uptake of such behaviors. However, low perceived effectiveness can dramatically undermine protective behaviors; this happened, for example, in Germany, with the introduction of the 70%-effective Astra Zeneca vaccine after other vaccines with 95% effectiveness had been introduced before. Emphasizing the benefits for others, for society, may lower these detrimental effects. When new behaviors are introduced, it may be worthwhile to create artificial past experiences, e.g., by building on social learning or exploring the feasibility of virtual reality technology to simulate experiences.

## Supporting information

**S1 File. Supplementary tables and figures, and sensitivity analysis.** S1 Table: Descriptive Statistics per country; S2 Table: Sample quotas per country; S3 Table: Models 1–2 replicated with experimental condition; S4 Table: Models 3–4 replicated with experimental condition; S5 Table: Model 1 per Country; S6 Table: Model 2 per Country; S7 Table: Model 3 per Country; S8 Table: Model 4 per Country; S1 Fig: Correlation Matrix; Sensitivity analysis.
(DOCX)

## Author Contributions

**Conceptualization:** Ana Paula Santana, Lars Korn, Cornelia Betsch, Robert Böhm.

**Data curation:** Ana Paula Santana.

**Formal analysis:** Ana Paula Santana, Lars Korn.

**Funding acquisition:** Cornelia Betsch, Robert Böhm.

**Investigation:** Ana Paula Santana, Lars Korn, Cornelia Betsch, Robert Böhm.

**Methodology:** Ana Paula Santana, Lars Korn, Cornelia Betsch, Robert Böhm.

**Resources:** Cornelia Betsch.

**Supervision:** Robert Böhm.

**Visualization:** Ana Paula Santana.

**Writing – original draft:** Ana Paula Santana.

**Writing – review & editing:** Ana Paula Santana, Lars Korn, Cornelia Betsch, Robert Böhm.

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
