## [Decision Letter · Decision Letter 0]

20 Jan 2022

PONE-D-21-31609Lessons learned about willingness to adopt various protective measures during the early COVID-19 pandemic in three countriesPLOS ONE

Dear Dr. Souza Santana,

Thank you for submitting your manuscript to PLOS ONE. After careful consideration, we feel that it has merit but does not fully meet PLOS ONE’s publication criteria as it currently stands. Therefore, we invite you to submit a revised version of the manuscript that addresses the points raised during the review process.

It seems that the Reviewers indicated several important issues. Please try to repond to them and accordingly adjust your manuscript.

We look forward to receiving your revised manuscript.

Kind regards,

Mariusz Duplaga, Ph.D., M.D., Ass. Prof.

Academic Editor

PLOS ONE

Journal Requirements:

2. In order to improve reporting, in your methods section, please provide additional information about the participant recruitment method.

3. Thank you for stating the following in the Acknowledgments/ Funding Section of your manuscript: 

The work was funded by grants from the German Research Foundation (DFG) to CB (BE 3970/8-2, BE 3970/11-1, BE 3970/12-1) and RB (BO-4466/2-2). The funding source did not influence the design of the study or the analysis of the results.

The work was funded by grants from the German Research Foundation (DFG, https://www.dfg.de/en/) to CB (BE 3970/8-2, BE 3970/11-1, BE 3970/12-1) and RB (BO-4466/2-2). 

Reviewers' comments:

Reviewer's Responses to Questions

**Comments to the Author**

1. Is the manuscript technically sound, and do the data support the conclusions?

Reviewer #1: Yes

Reviewer #2: Partly

2. Has the statistical analysis been performed appropriately and rigorously? 

Reviewer #1: Yes

Reviewer #2: Yes

3. Have the authors made all data underlying the findings in their manuscript fully available?

Reviewer #1: Yes

Reviewer #2: Yes

4. Is the manuscript presented in an intelligible fashion and written in standard English?

Reviewer #1: Yes

Reviewer #2: Yes

5. Review Comments to the Author

Reviewer #1: The manuscript presents a study conducted on representative samples from three countries (Germany, Hong Kong, US). A study aimed to verify three hypotheses about factors predicting intentions towards COVID-19 preventive behaviors. The study is methodologically correct, and the manuscript is well-written. I especially acknowledge that the authors followed Open Science recommendations such as preregistration, transparency of the description of methods and results (e.g., the distinction between preregistered and exploratory analyses).

I have only minor comments that may help to improve the manuscript.

1. I recommend the authors share the exact wording of questions used in the study (in English) to help researchers in the future replicate the study on the OSF project.

2. Please highlight in the manuscript that the codes of hypotheses (e.g., H1, H2, H3) are not consistent with the preregistration (H4, H5, H6)

3. Please elaborate more on psychological mechanisms that may underlie the effectiveness and experience hypotheses. How may perceived effectiveness shape self-efficacy/controllability/motivation for a particular behavior? Whether the experience in performing specific action only helps shape habits or maybe you can find an alternative explanation, e.g., people may want to shape a consistent image of themselves? Moreover, perhaps it would be good to include risk perception as a concept explaining some of these relationships (even if you decided not to have it in the analyses reported in this manuscript)?

4. Please briefly discuss how not following the exclusion criterion (i.e., being infected with COVID-19) could influence results.

5. Please provide in the manuscript sample items of individualism and collectivism. Please briefly explain what horizontal and vertical individualism are.

6. Why did you ask participants about washing hands for 20 seconds while in the introduction you argue that it is recommended to wash hands for 30 seconds?

7. I recommend adding (e.g., in supplementary materials) the correlation matrix between the most important variables used in the study)

8. The results showed a very high willingness to adopt the measures used in the study (M 5.6.-6.10/7 point scale). Please stress this as a limitation of a study and discuss how the pattern of results may be found in other (less compliant) countries.

9. Please comment about the relative strength of predictive power for the most important variables

Reviewer #2: Thank you for giving me the opportunity to review this interesting paper. In this research, the willingness to adopt specific protective measures against COVID-19 was assessed in three countries. There are many strengths to this research, including preregistration of the study and the provision of all data, materials, and code. Beside the strengths of this research, there are also some concerns that need to be addressed. The major concerns of this research are described in the following:

1. According to the preregistration, this paper describes the second (or additional) part of an experimental study. In the first part, participants received information about a hypothetical vaccine; the vaccine effectiveness was either 51% or 75% (between-subjects design). Even though the variables analysed for this paper were probably an addendum to the experiment, it is possible that the different experimental manipulations had an effect on the variables measured in the second part. The authors should be more open about the fact that the first part of the study was an experiment, which is not clear from the current description of the study (e.g., on page 8, the word ‘experiment’ is not mentioned). In addition, all analyses should include the experimental condition as an additional factor to determine whether the experimental manipulation had an (unintended) effect on the variables presented here.

2. The theoretical framework, which gives a basis for hypotheses and choice of research methods, could be described in more detail. It is not clear to me why effectiveness, previous experience, and intended self-and other-protection were chosen as the core predictors, whereas perceived risks, norms and trust were not included in the analyses. The authors could justify their selection a bit more and refer to models such as Theory of Planned Behavior or Protection Motivation Theory.

3. Please provide more details on participant screening and sampling. According to the preregistration, participants who did not fully completed the study or who reported to have an infection with COVID-19 were excluded. This is not mentioned in the manuscript. In the current version of the manuscript, there is also no information about the drop-out during the study, i.e. how many participants did not finish the study although they fulfilled the quota requirements. Please also provide more information about the panel provider and the compensation participants received for participation.

4. The authors state that the “aim of sampling participants from several countries was not to compare responses between countries, but rather to create sufficient variance in the predictors of interest, which have been shown to vary between countries and cultural contexts” (page 8). I assume that a sample from Hong Kong was included because Hong Kong is a collectivist culture. However, Table S2 indicates that compared to the other two countries, the sample form Hong Kong scored lowest on collectivism and highest on individualism. Presumably, therefore, a bias has arisen due to the type of recruitment/sampling. This should be emphasized more strongly and reasons for this should be discussed in more detail. It would also be helpful if Table S2 could be supplemented with information on significant differences between the countries.

There were also some other (minor) concerns:

• Abstract (last sentence). Because the results of the experiment are not presented in this manuscript, I wonder why the last sentence addresses communication of vaccine effectiveness. Perhaps a different conclusion can be chosen here.

• Please keep verb tense consistent in sentences and paragraphs (e.g. page 10, line 221).

• Please keep citation style consistent (e.g., page 11, line 242).

• Figure 2 is a bit small. It is difficult to grasp which variable labels belong to which coefficients. Maybe you can change the figure so that the variable labels are only shown once and the coefficients from the three different samples are shown in different colours. This would also facilitate the comparison of the results of the different countries.

6. PLOS authors have the option to publish the peer review history of their article (what does this mean?). If published, this will include your full peer review and any attached files.

Reviewer #1: **Yes: **Agata Sobkow

Reviewer #2: No

---

## [Author Response · Author response to Decision Letter 0]

21 Feb 2022

The pages referred to here correspond to the marked-up manuscript document.

Editor

Editor, Comment 1: Please ensure that your manuscript meets PLOS ONE's style requirements, including those for file naming. The PLOS ONE style templates can be found at https://journals.plos.org/plosone/s/file?id=wjVg/PLOSOne_formatting_sample_main_body.pdf and 

We have now adapted the manuscript and file names according to the PLOS ONE style templates.

Editor, Comment 2: In order to improve reporting, in your methods section, please provide additional information about the participant recruitment method.

More information about participants and recruitment has been added to the respective section: “Participants were from Germany (n = 333), Hong Kong (n = 367), and the United States (US) (n = 495) and were members of an ISO 26362:2009-compliant online panel (37). The external panel provider was responsible for recruitment and compensation of the participants. Due to problems in filling certain quotas, the U.S. sample was oversampled to reach the required quotas. Drop-out rates after quota filling were 7.92% for Germany, 11.16% for Hong Kong, and 15.05% for the United States. The aforementioned sample sizes only consider complete participation.” (p. 9)

Editor, Comment 3: Thank you for stating the following in the Acknowledgments/ Funding Section of your manuscript: (…). Please include your amended statements within your cover letter; we will change the online submission form on your behalf. 

We have now excluded the funding information from the manuscript and included our amended statement in the cover letter.

Reviewer 1

Reviewer 1, Comment 1: The manuscript presents a study conducted on representative samples from three countries (Germany, Hong Kong, US). A study aimed to verify three hypotheses about factors predicting intentions towards COVID-19 preventive behaviors. The study is methodologically correct, and the manuscript is well-written. I especially acknowledge that the authors followed Open Science recommendations such as preregistration, transparency of the description of methods and results (e.g., the distinction between preregistered and exploratory analyses).

I have only minor comments that may help to improve the manuscript.

We thank this reviewer for the overall positive feedback.

Reviewer 1, Comment 2: I recommend the authors share the exact wording of questions used in the study (in English) to help researchers in the future replicate the study on the OSF project.

We added the exact wording of all study materials to the OSF supplement.

Reviewer 1, Comment 3: Please highlight in the manuscript that the codes of hypotheses (e.g., H1, H2, H3) are not consistent with the preregistration (H4, H5, H6)

We added a footnote to highlight this: “1Note that the numbering of the hypotheses reported here differs from the pre-registration, where they are numbered as H4, H5, and H6, respectively.” (p. 5).

Reviewer 1, Comment 4: Please elaborate more on psychological mechanisms that may underlie the effectiveness and experience hypotheses. How may perceived effectiveness shape self-efficacy/controllability/motivation for a particular behavior? Whether the experience in performing specific action only helps shape habits or maybe you can find an alternative explanation, e.g., people may want to shape a consistent image of themselves? Moreover, perhaps it would be good to include risk perception as a concept explaining some of these relationships (even if you decided not to have it in the analyses reported in this manuscript)?

Regarding the effectiveness hypothesis, we elaborated on the relevance of perceived effectiveness to increase motivation to adopt the behavior, pointing out that this is an important construct for different theoretical frameworks (p. 4/5). Regarding the experience hypothesis, we now refer to the concepts of habit formation and learning from past choices (p. 5/6).

Reviewer 1, Comment 5: Please briefly discuss how not following the exclusion criterion (i.e., being infected with COVID-19) could influence results.

We now discuss this issue: “Another limitation arises from the fact that we could not comply with one of the pre-registered exclusion criteria (i.e., whether one has been infected with COVID-19), and this might have affected our results. Overall, individuals who had experience with the disease, and depending on the type of experience (e.g., mild or severe), could have adapted their risk perceptions and health behaviors accordingly. For instance, if someone was infected and had only mild symptoms, they could have perceived the disease as posing less risk, subsequently reducing their protective behavior. As the data collection took part at a very early time during the pandemic, however, we assume that only a very small proportion might have had been infected at all.” (p. 22/23)

Reviewer 1, Comment 6: Please provide in the manuscript sample items of individualism and collectivism. Please briefly explain what horizontal and vertical individualism are.

Following the reviewer’s suggestion, we have added sample items for both scales in the methods section. Although the used scale makes the distinction between horizontal and vertical individualism/collectivism, we did not make such differentiation in our study. Instead, we used both horizontal and vertical subscales of collectivism to compute the average score for this construct. We now explicitly note this in the respective section:

“We did not differentiate between the horizontal and vertical subtypes of the subscales. Thus, only two scores (i.e., individualism and collectivism) were computed by the average of items of each dimension (Cronbach’s α was 0.73 for individualism and 0.84 for collectivism). Sample items are: “I’d rather depend on myself than others.”, for the individualism subscale; and “The well-being of my co-workers is important to me.”, for the collectivism subscale.” (p. 10)

Reviewer 1, Comment 7: Why did you ask participants about washing hands for 20 seconds while in the introduction you argue that it is recommended to wash hands for 30 seconds?

The article cited in the referred paragraph indeed suggests washing hands for 30 seconds as a protective measure. However, we have now updated the information citing the Centers for Disease Control and Prevention (CDC), using the recommendation of washing for at least 20 seconds. This guideline is consistent across the three countries investigated.

Hong Kong:

https://www.chp.gov.hk/files/pdf/lts_covid-19_20200525_eng.pdf

United States:

https://www.cdc.gov/coronavirus/2019-ncov/prevent-getting-sick/prevention.html

Germany: https://assets.zusammengegencorona.de/eaae45wp4t29/59kagmqA8AWs87Jo2VgpTf/f826234fc4d8d58a891c9d159d2627aa/BMG_Infoflyer_kurz_Corona_English.pdf

Reviewer 1, Comment 8: I recommend adding (e.g., in supplementary materials) the correlation matrix between the most important variables used in the study)

We added a correlation matrix to the Supplementary Material (see Figure S1).

Reviewer 1, Comment 9: The results showed a very high willingness to adopt the measures used in the study (M 5.6.-6.10/7 point scale). Please stress this as a limitation of a study and discuss how the pattern of results may be found in other (less compliant) countries.

This limitation is now addressed: “Second, our sample showed a high willingness to adopt the behaviors, which might have resulted in an underestimation of the effects found. For instance, in countries where compliance and regulations are weaker, or at times when compliance wanes due to habituation effects, individual differences are likely to matter more (46, 47).” (p. 22)

Reviewer 1, Comment 10: Please comment about the relative strength of predictive power for the most important variables

We added the following to the discussion: “Considering the variables investigated here, our results also offer insights into the relative importance of the predictors. Perceived effectiveness as well as intended self- vs. other-protection were the strongest predictors of the willingness to adopt the behaviors. This trend is also reflected in the models using the PMI. Given that these variables were of particular importance, it could be fruitful to target them when aiming to increase compliance with protective measures.” (p. 22)

 

Reviewer 2

Reviewer 2, Comment 1: Thank you for giving me the opportunity to review this interesting paper. In this research, the willingness to adopt specific protective measures against COVID-19 was assessed in three countries. There are many strengths to this research, including preregistration of the study and the provision of all data, materials, and code. Beside the strengths of this research, there are also some concerns that need to be addressed. The major concerns of this research are described in the following:

We thank the reviewer for the overall positive feedback.

Reviewer 2, Comment 2: According to the preregistration, this paper describes the second (or additional) part of an experimental study. In the first part, participants received information about a hypothetical vaccine; the vaccine effectiveness was either 51% or 75% (between-subjects design). Even though the variables analysed for this paper were probably an addendum to the experiment, it is possible that the different experimental manipulations had an effect on the variables measured in the second part. The authors should be more open about the fact that the first part of the study was an experiment, which is not clear from the current description of the study (e.g., on page 8, the word ‘experiment’ is not mentioned). In addition, all analyses should include the experimental condition as an additional factor to determine whether the experimental manipulation had an (unintended) effect on the variables presented here.

This information has now been available: “The present data came from an independent survey included in a larger study, which was an experiment investigating the intention to get vaccinated with a hypothetical vaccine.” (p. 9)

Further, we added to the Methods and Materials section: “Considering this study was an addition to an experimental study, we also conducted additional analyses including the experimental condition as predictor. Although most predictors were measured after the experimental manipulation, it did not affect the measures. The corresponding analyses are reported in the Additional File 1 (Tables S3-S4).” (p. 11/12)

Reviewer 2, Comment 3: The theoretical framework, which gives a basis for hypotheses and choice of research methods, could be described in more detail. It is not clear to me why effectiveness, previous experience, and intended self-and other-protection were chosen as the core predictors, whereas perceived risks, norms and trust were not included in the analyses. The authors could justify their selection a bit more and refer to models such as Theory of Planned Behavior or Protection Motivation Theory.

Following the reviewer’s suggestion, as well as Comment 4 from Reviewer 1, we now provide more evidence explaining the potential mechanisms behind the effectiveness hypothesis (p. 4/5) and experience hypothesis (p. 5/6). 

Reviewer 2, Comment 4: Please provide more details on participant screening and sampling. According to the preregistration, participants who did not fully completed the study or who reported to have an infection with COVID-19 were excluded. This is not mentioned in the manuscript. In the current version of the manuscript, there is also no information about the drop-out during the study, i.e. how many participants did not finish the study although they fulfilled the quota requirements. Please also provide more information about the panel provider and the compensation participants received for participation.

As mentioned in the Methods and Materials section, due to technical issues, the question about being infected with COVID-19 was not included in the study. Therefore, we could not exclude based on this pre-registered item. 

The following information regarding the sampling procedure was added/changed: “The sample of participants (N = 1,195; Mage = 47.56, SD = 17.41; 46.53% female) was recruited online between 04/24/2020 and 05/01/2020. Participants were from Germany (n = 333), Hong Kong (n = 367), and the United States (US) (n = 495) and were members of an ISO 26362:2009-compliant online panel. The external panel provider used was responsible for recruitment and compensation of the participants. Due to problems in filling certain quotas, the U.S. sample was oversampled to reach the required quotas. Drop-out rates after quota filling were 7.92% for Germany, 11.16% for Hong Kong, and 15.05% for the United States. The aforementioned sample sizes only consider complete participation.” (p. 9)

Reviewer 2, Comment 5: The authors state that the “aim of sampling participants from several countries was not to compare responses between countries, but rather to create sufficient variance in the predictors of interest, which have been shown to vary between countries and cultural contexts” (page 8). I assume that a sample from Hong Kong was included because Hong Kong is a collectivist culture. However, Table S2 indicates that compared to the other two countries, the sample form Hong Kong scored lowest on collectivism and highest on individualism. Presumably, therefore, a bias has arisen due to the type of recruitment/sampling. This should be emphasized more strongly and reasons for this should be discussed in more detail. It would also be helpful if Table S2 could be supplemented with information on significant differences between the countries.

We thank the reviewer for pointing this out. We have now added information on significant differences between countries in the Additional File (Table S1). Although countries significantly differed in levels of collectivism, they did not differ in levels of individualism. However, we would like to highlight that we used individual scores of collectivism and individualism in our analyses and, therefore, our results are based on individual-level variance. Therefore, we believe it’s rather secondary whether there were differences between countries, as this is not the focus of the analyses. 

Nevertheless, following the reviewer’s suggestion, we now explicitly mention the somewhat unusual country-level scores in the discussion: “Finally, there were only small differences in the levels of collectivism and individualism between countries, and, surprisingly, the average level of collectivism (individualism) of participants from Hong Kong was descriptively below (above) the levels from German and U.S. participants (see Table S1). We refrain from speculations why this was the case but want to highlight that the analyses and their interpretations are based on individual-level responses instead.” (p. 23)

Reviewer 2, Comment 6: Abstract (last sentence). Because the results of the experiment are not presented in this manuscript, I wonder why the last sentence addresses communication of vaccine effectiveness. Perhaps a different conclusion can be chosen here. 

The sentence about vaccine effectiveness was removed.

Reviewer 2, Comment 7: Please keep verb tense consistent in sentences and paragraphs (e.g. page 10, line 221). 

We have now changed it accordingly.

Reviewer 2, Comment 8: Please keep citation style consistent (e.g., page 11, line 242). 

This mistake was corrected.

Reviewer 2, Comment 9: Figure 2 is a bit small. It is difficult to grasp which variable labels belong to which coefficients. Maybe you can change the figure so that the variable labels are only shown once and the coefficients from the three different samples are shown in different colours. This would also facilitate the comparison of the results of the different countries. 

As suggested, we updated the picture to the following:

(see Response to Reviewers document)

---

## [Decision Letter · Decision Letter 1]

10 Mar 2022

Lessons learned about willingness to adopt various protective measures during the early COVID-19 pandemic in three countries

PONE-D-21-31609R1

Dear Dr. Souza Santana,

We’re pleased to inform you that your manuscript has been judged scientifically suitable for publication and will be formally accepted for publication once it meets all outstanding technical requirements.

Kind regards,

Mariusz Duplaga, Ph.D., M.D., Ass. Prof.

Academic Editor

PLOS ONE

Additional Editor Comments (optional):

Reviewers' comments:

Reviewer's Responses to Questions

**Comments to the Author**

1. If the authors have adequately addressed your comments raised in a previous round of review and you feel that this manuscript is now acceptable for publication, you may indicate that here to bypass the “Comments to the Author” section, enter your conflict of interest statement in the “Confidential to Editor” section, and submit your "Accept" recommendation.

Reviewer #1: All comments have been addressed

Reviewer #2: All comments have been addressed

2. Is the manuscript technically sound, and do the data support the conclusions?

Reviewer #1: Yes

Reviewer #2: Yes

3. Has the statistical analysis been performed appropriately and rigorously? 

Reviewer #1: Yes

Reviewer #2: Yes

4. Have the authors made all data underlying the findings in their manuscript fully available?

Reviewer #1: Yes

Reviewer #2: Yes

5. Is the manuscript presented in an intelligible fashion and written in standard English?

Reviewer #1: Yes

Reviewer #2: Yes

6. Review Comments to the Author

Reviewer #1: (No Response)

Reviewer #2: The authors have been responsive to my comments. I can now recommend the publication of this manuscript.

7. PLOS authors have the option to publish the peer review history of their article (what does this mean?). If published, this will include your full peer review and any attached files.

Reviewer #1: No

Reviewer #2: No

---

## [Editor Report · Acceptance letter]

16 Mar 2022

PONE-D-21-31609R1 

Lessons learned about willingness to adopt various protective measures during the early COVID-19 pandemic in three countries 

Dear Dr. Souza Santana:

I'm pleased to inform you that your manuscript has been deemed suitable for publication in PLOS ONE. Congratulations! Your manuscript is now with our production department. 

Kind regards, 

on behalf of

Dr. Mariusz Duplaga 

Academic Editor

PLOS ONE